# VERY FAST GRAPH CLUSTERING FOR SINGLE AND MULTIPLE VIEWS

## ABSTRACT

Clustering is a fundamental step in learning and analyzing graphs. Commonly accepted criteria for evaluating graph clustering quality without ground truth are the "normalized cut" (ncut), and the "ratio cut" (rcut). Traditional algorithms that minimize ncut and rcut take $O(mnk)$ to cluster a graph of $n$ nodes and $m$ edges into $k$ clusters. Faster algorithms sacrifice accuracy for speed and run in $O(m+nk^2)$. A very recent algorithm runs in $O(m+nk\log k)$. The space complexity of these algorithms ranges from $O(n^2)$ to $O(n\log k)$. We describe a new algorithm with running time of $O(m\log m)$ that achieves accuracy similar to traditional algorithms. Our algorithm is simple to implement, and requires only $O(m)$ memory. It can also be applied in the multi-view setting, where multiple graphs share the same set of nodes. Our algorithm can cluster a small number of views with no increase in its running time. We describe a randomized implementation that allows a qualitative comparison between various internal clustering criteria. Our experiments suggest a new criterion that we call "linfcut" as superior to both the ncut and the Cheeger criteria, computing clusters that "make more sense" to a human observer. Our algorithm performs a search for edges between clusters. Its speed is the result of a strong "ignorance" (pruning) condition that allows ignoring most of the edges after little computation.

## 1 INTRODUCTION

Graph representation of data provides information about the relationship between data items. Clustering the graph nodes is a fundamental step in graph learning, that was very heavily studied. A successful general approach is *spectral clustering*, which relies on eigenvectors of the graph Laplacian. For general references see, e.g. Meila (2016); Bolla (2013); Nascimento & Carvalho (2011); von Luxburg (2007).

Closely related to graph clustering is the problem of *multi-view graph clustering*. Here it is assumed that there are multiple graphs sharing the same set of nodes, each describing a distinct relationship between the data items. Multi-view graph clustering searches for a node partition that provides a good clustering for each one of the multiple graphs. For general references see, e.g. Fang et al. (2023); Chao et al. (2021).

### 1.1 THE CLASSICAL ALGORITHM FOR $k$ CLUSTERING

Spectral clustering was popularized by the work of Shi & Malik (2000). They computed $k$ clusters by iteratively partitioning a single cluster into a pair of clusters. See von Luxburg (2007) for additional details. The main theme that we follow started with Ng et al. (2001), where $k$ eigenvectors were used to obtain $k$ clusters.

Let $G$ be an undirected graph of $n$ nodes and $m$ edges. Let $w_{ij}$ be the non-negative weight between nodes $i$ and $j$, so that the matrix $W=(w_{ij})$ is $n\times n$. The degree of node $i$ is $d_i=\sum_j w_{ij}$. The degree vector is $d=(d_1,\ldots,d_n)^T$, and the $n\times n$ degree matrix is $D=\mathsf{diag}(d)$. The graph Laplacian is $L=D-W$. The Laplaian can be normalized in various ways, giving rise to different clustering criteria. We denote the normalization matrix by $P=\mathsf{diag}(p_1,\ldots,p_n)$. It is a diagonal matrix with

the diagonal elements $p_i > 0$. The normalized Laplacian $\tilde{L}$ is defined by the following formula:

$$\tilde{L} = P^{-1/2} L P^{-1/2}.$$

The most common choices of $P$ are given in the following definition:

**Definition 1.** *Normalization matrices.*

**Ratio Cut Normalization:** *Here $p_i = 1$, so that $P = I$, and $\tilde{L} = L$. See von Luxburg (2007).*
**Normalized Cut Normalization:** *Here $p_i = d_i$, $P = D$, and $\tilde{L} = D^{-1/2} L D^{-1/2}$.*
     *See von Luxburg (2007).*

The related clustering criteria are specified in the following definition:

**Definition 2.** *Suppose the $n$ nodes of $G$ are partitioned into the $k$ disjoint partitions $A_1, \ldots, A_k$. Let $P = \mathrm{diag}(p_1, \ldots, p_n)$ be a normalization matrix.*
*The **volume** of partition $t$ is defined by: $V_t = \sum_{i \in A_t} p_i$.*
*The **cut** of partition $t$ is defined by: $C_t = \sum w_{i,j}$, where either $i$ or $j$ are in $A_t$ but not both.*
*The **generalized normalized cut** of the partition $A_1, \ldots, A_k$ is: $\mathsf{gcut} = \frac{1}{2} \sum_{t=1}^{k} \frac{C_t}{V_t}$.*
*In particular, **Ratio Cut**, and **Normalized Cut**, are obtained from the normalizations in Definition 1.*

---

**Algorithm 1:** Classical Spectral Clustering

---

**Input:** A graph $G$ with $n$ vertices and $m$ nodes to be clustered into $k$ clusters. A choice of $P$.
**Output:** The $k$ clusters.
1. Compute the normalized Laplacian $\tilde{L} = P^{-1/2} L P^{-1/2}$.
2. Compute $u_0, \ldots, u_{k-1}$, the $k$ eigenvectors of $\tilde{L}$ corresponding to its $k$ smallest eigenvalues. Set $U = (u_0, \ldots, u_{k-1})$. $U$ is $n \times k$. Normalize each row of $U$ to a unit vector.
3. Consider the nomalized rows of $U$ as $n$ points in $R^k$. Use the $k$-means algorithm to cluster these points into $k$ clusters. Return these $k$ clusters as output.

---

The reason that Algorithm 1 works is that it minimizes normalized cuts. The proof of this fact relies on the following theorem:

**Theorem 1.** *Let $A_1, \ldots, A_k$ be a disjoint partition of the graph nodes. Then there are $k$ cluster indicator vectors $q_1, \ldots, q_k$ such that:*

$$\mathsf{gcut}(A_1, \ldots, A_k) = \sum_{j=1}^{k} \frac{q_j^T \tilde{L} q_j}{q_j^T q_j} \tag{1}$$

For the proof see von Luxburg (2007).

Indicator vectors have the same value for coordinates inside the cluster, and another value for coordinates outside the cluster. Minimizing Equation (1) for such discrete values is NP-hard (e.g., Shi & Malik (2000)), but if it is relaxed to real values, then from the Courant Fischer theorem the global minimum is obtained with the $k$ eigenvectors used in Algorithm 1. Therefore, the eigenvectors computed in Algorithm 1 should be approximate linear combinations of indicator vectors, which indicates that they should have approximately the same value over each cluster. In such a case the clustering should be recoverable by $k$-means.

In summary, the classical spectral clustering algorithm performs well minimizing normalized cuts because it finds the globally optimal solution of a relaxed problem.

## 1.2 PREVIOUS STUDIES

Algorithm 1 is easy to implement since eigenvalue routines and $k$-means implementations are readily available. However, direct implementations may be very slow when applied to large graphs. Much of the research in this area, including our result, is aimed at improving the running time.

Table 1: Complexity of several spectral clustering algorithms

| Line | Citations | Running time | Memory | Notes |
|---|---|---|---|---|
| 1 | Ng et al. (2001) | $O(nmk)$ | $O(n^2)$ | |
| 2 | Fowlkes et al. (2004) | $O(m+nlk+l^3)$ | $O(nl)$ | $l \geq k$ |
| 3 | Li et al. (2011) | $O(m+nlk)$ | $O(nk)$ | $l \geq k$ |
| 4 | Cai & Chen (2015) | $O(m+nl^2)$ | $O(nk)$ | $l \geq k$ |
| 5 | Wang et al. (2021) | $O(m + nlk)$ | $O(n^2)$ | $l \geq k$ |
| 6 | Hai Zhang & Chang (2022) | $O(n^2l)$ | $O(nl)$ | $l \geq k$ |
| 7 | Chen et al. (2023) | $O(m+nk^2)$ | $O(nk)$ | |
| 8 | Macgregor (2023) | $O(m+nk \log k)$ | $O(n \log k)$ | |
| 9 | Our Algorithm | $O(m \log m)$ / $O(m \log m \log n)$ | $O(m)$ | |
| 10 | Our Algorithm, multiview | $O(m \log m+v(n-k) \log n)$ | $O(m)$ | |

### 1.2.1 SINGLE VIEW SPECTRAL CLUSTERING

Deep learning approaches to spectral clustering attempt to infer the graph from the data, and combine this with steps 1,2 of the algorithm. Examples of this approach include SpectralNet of Shaham et al. (2018), DivClust of Metaxas et al. (2023), and the dual aoutoencoder network of Yang et al. (2019). See also the overview by Wei et al. (2024).

These deep learning approaches cannot be directly compared to the conventional approaches because they have a different input. In terms of running time they do better than the conventional "old" algorithms (see Table 1), but do not seem to be competitive with the more recent fast clustering methods described in this section,

There is a very large number of studies on various aspects of traditional spectral clustering. We are mostly interested in the running time, which is the focus of most current research. Table 1 lists several algorithms with their time and space complexity. With the exception of our algorithm, all others can be viewed as modifications of Algorithm 1. The older algorithms are dominated by the complexity of computing the $k$ eigenvectors, while the running time of the newer algorithms approaches the running time of the $k$-means step.

Line 1 of Table 1 corresponds to he original spectral clustering algorithm. With an efficient eigenvalue routine its running time is $O(nmk)$, which is considered impractical for many applications. The algorithms cited in lines 2, 3 select $l$ random columns and approximate eigenvectors from the $n \times l$ sample matrix. A different sampling technique is used with the algorithm cited in Line 4, which uses $k$-means (with large $k$) to find the sampled columns. These and other algorithms that follow similar ideas can be found in a review paper by Tremblay & Loukas (2020). The paper cited in Line 5 suggests an alternative to $k$ means, which can be implemented with roughly the same cost. The papers cited on lines 6,7 apply recently developed randomized techniques to compute the eigenvectors.

A very recent algorithm, cited in Line 8, comes with an even lower running time. The main idea is to use only $\log k$ eigenvectors (or other equivalent vectors). This reduces the complexity of the $k$-means step to $O(nk \log k)$, which dominates the complexity of the algorithm.

The complexity of the algorithm we propose is shown in Line 9. Its running time is data dependent. The worst case running time is $O(m \log m \log n)$, but the expected time is $O(m \log m)$, For sparse graphs, where $m=O(n)$, our algorithm is much faster than the others, especially for large $k$ values. Even when compared to Macregor's algorithm shown in Line 8, our algorithm is faster for large $k$ and sparse graphs. A detailed experimental comparison is given in Section 3.

### 1.2.2 MULTI-VIEW CLUSTERING

When given multiple graphs as distinct views of the data, multi-view clustering attempts to discover node clustering that is good "on the average", where the average is taken over the multiple views. The traditional approaches attempt to compute averages of properties that can be extracted from the single view graphs, and then compute the clustering from these averages. For example, the "Co-regularized Multi-view Spectral Clustering" algorithm of Kumar et al. (2011) attempts to

average cluster indicator matrices computed from the views. Zhong & Pun (2022) computes relaxed clustering from each view and then attempts to average them to obtain a single view.

Unlike the single view case, multi-view clustering algorithms do not solve optimally a relaxed continuous problem. We point out that the typical running time of current multi-view techniques can be significantly higher than a similar single-view problem. By contrast, the algorithm we propose has roughly the same complexity for both single and multiple views.

### 1.3 OUR APPROACH

The current state of the art spectral clustering algorithms may be too slow for "massive graphs" that appear in many current applications. See Bader (2022) for a variety of practical problems involving graphs with billions of nodes. Our goal in this study is to improve the current state of the art in terms of speed, without significantly reducing accuracy. We describe a very fast clustering algorithm for minimizing ratio criteria (see Definition 2), which can be directly compared with traditional spectral clustering algorithms.

**The basic algorithm.** We propose a greedy procedure for minimizing cuts, showing how to obtain $k$-clustering from $k+1$-clustering. Applying this recursively ($n-k$ times) produces $k$-clustering from the initial (trivial) $n$-clustering. The same approach can be used, with minimal changes, in the multi-view case.

**Improving the speed.** The greedy approach described above is not fast enough, resulting in algorithms with running time proportional to $mn$. Examining Table 1 it is clear that this running time is not competitive. Our main result is a reduction of this running time to $O(m \log m)$ on the average, $O(m \log m \log n)$ worst case. The key idea is the observation that large portions of the calculations in the greedy algorithm need not be carried out at all. We note that for sparse graphs $O(m \log m \log n)$ running time is faster than a single iteration of $k$-means for sufficiently large $k$.

**Randomization.** The next issue that we investigate is how to randomize the algorithm. The randomization produces an algorithm that is no longer greedy. The challenge that we solve in Section 2.3 is how to obtain such randomization without significantly reducing the running time. The randomization improves the algorithm accuracy and allows several extensions.

**Several top solutions.** A simple extension of the randomized algorithm produces as output several top solutions that were obtained during the multiple runs. For a different method of producing multiple solutions see, e.g. Guedes et al. (2016).

**Cheeger and linfcut criteria.** The randomized algorithm enables evaluating various clustering criteria. We describe experiments with the generalized Cheeger criterion e.g., Lee et al. (2014), and propose a new criterion that we call linfcut.

We summarize below our main contributions:

- An $O(m \log m)$ (expected) clustering algorithms that minimizes the same criteria as classical spectral clustering algorithms.
- An $O(m \log m)$ (expected) multi-view spectral clustering algorithm.
- Randomized variants of these algorithms that can produce several top solutions.
- Randomized variants of the above algorithms that can optimize various clustering criteria.
- The discovery of linfcut, a new clustering criterion that appears to have an advantage over previously proposed criteria, as judged by a human observer.

## 2 THE PROPOSED ALGORITHM

Recall Definition 2 where the generalized cut of the graph partition $A_1, \ldots, A_k$ is computed by a sum over the partition subsets. It can also be expressed by a sum over all edges as follows:

$$\mathsf{gcut} = \frac{1}{2} \sum_{t=1}^{k} \frac{C_t}{V_t} = \sum_{\text{edge } (i,j,w)} h(i,j,w), \text{ where: } h(i,j,w) = \begin{cases} 0 & A(i) = A(j) \\ w_{ij}\left(\frac{1}{V(i)} + \frac{1}{V(j)}\right) & A(i) \neq A(j) \end{cases} \quad (2)$$

Here $A(i)$ is the partition of $i$, and $V(i)$ is the volume of $A(i)$.

Suppose we are given a partition into $k+1$ subsets. It can be reduced to a partition of $k$ subsets by merging two subsets. Examining Equation (2) suggests the following simple greedy heuristic:

**Heuristic 1:** Merge the subsets connected by $i$ and $j$ that maximize $h(i, j, w)$.

Heuristic 1 can be used to design a $k$-clustering algorithm. The algorithm starts with the $n$ clustering obtained by considering each node as a cluster, and applies the heuristic $n-k$ times. It is shown below as Algorithm 2.

---

**Algorithm 2:** A greedy algorithm for $k$-clustering

---

**Input:** The $m$ edges $(i, j, w_{ij})$. The normalization values $p_1, \ldots, p_n$. (See Definition 1.)
**Output:** The $k$ clusters.
**Initialization:** Create the initial clustering as the $n$ subsets $A_1^0, \ldots, A_n^0$, where $A_i^0 = \{i\}$.
        Set the initial cluster volumes $V_1^0, \ldots, V_n^0$ as: $V_i^0 = p_i$.
**Iteration:** For $t = 0, \ldots, n-k-1$:
        **1.** Compute the value $h(i, j, w_{ij})$ for all edges using Equation (2).
        **2.** Select the edge $(i^*, j^*, w^*)$ such that: $h(i^*, j^*, w^*) = \max_{i,j} h(i, j, w)$.
        **3.** Remove the subsets $A^t(i^*)$, $A^t(j^*)$, replacing then with the subset:
            $A_*^t = A^t(i^*) \cup A^t(j^*)$, and set $V_*^t = V^t(i^*) + V^t(j^*)$.
        **4.** Rename the subsets $A_1^{t+1} \ldots, A_{n-(t+1)}^{t+1}$, and their corresponding volumes.
**Termination:** Produce as output the subsets $A_1^{n-k}, \ldots A_k^{n-k}$.

---

In the algorithm description we write $A(i)$ for the subset that contains $i$. This is typically called a "**find**" operation. The cost of naive implementations of the "**union**" in Step 3, and the "**find**" operations in Step 1 may overwhelm the algorithm complexity. However, using the classical "**Union-Find**" data structure reduces the cost of each "**Union**" to $\log n$, and the cost of each "**find**" operation to (almost) a constant. See, e.g. Cormen et al. (1992). With this implementation the complexity of Algorithm 2 is $O(m(n-k))$. Still, according to Table 1 this is not competitive.

We proceed to describe our main result, a fast implementation of Algorithm 2. This is accomplished using a binary heap. Recall that max-heap is a data structure that provides fast implementation of the following operations (among others): **Make_Heap** - $O(m)$, **Insert_Element** - $O(\log m)$, **View_Max** - $O(1)$, **Extract_Max** - $O(\log m)$. The implementation is shown in Algorithm 3.

---

**Algorithm 3:** Fast graph clustering

---

**Input:** The $m$ edges $(i, j, w_{ij})$. The normalization values $p_1, \ldots, p_n$. (See Definition 1.) $k$ - the
        desired number of clusters.
**Output:** The clusters.
**Initialization:** Create the initial clustering as the $n$ subsets $A_1, \ldots, A_n$, where $A_i = \{i\}$.
        Set the corresponding volumes $V_1, \ldots, V_n$. $V_i = p_i$.
        Insert the subsets and their volumes into a **Union-Find**.
        Compute the value $h(i, j, w_{ij})$ for all edges using Equation (2).
        **Make_Heap** from the edges with their corresponding $h$ values.
**Iteration:** Repeat until $n-k$ edges are selected:
        **1.** **Extract_Max** to obtain the edge $e_1 = (i_1, j_1, w_1)$ with the value $h_{1\_old}$
        **2.** Update the value of $h_{1\_old}$ using Equation (2) with the most current clustering
            information in **Union-Find**. Denote the new value $h_{1\_new}$.
        **3.** **View_Max** on the heap (after $e_1$ extraction) to obtain the edge $e_2$ with value $h_{2\_old}$.
        **4.** If $h_{1\_new} \geq h_{2\_old}$ perform **Union**$(A(i_1), A(j_1))$. Otherwise
            **Insert_Element**$(e_1, h_{1\_new})$.
**Termination:** The output is the subsets in the **Union-Find** data structure.

---

## 2.1 ANALYSIS OF ALGORITHM 3

An edge can be extracted (in Line 1 of the Iteration), then updated (in Line 2), and then re-inserted (in Line 4). An important observation is that the edge value cannot increase in this cycle.

**Proposition 1.** *The value associated with an edge in the heap cannot increase.*

**Proof:** *According to Equation (2) the value of $h(i, j, w)$ is inversely related to $V(i), V(j)$. The proof follows from the observation that during the run of the algorithm these volumes can only grow, but never shrink.* □

Observe that an edge is only evaluated after it is extracted in Line 1 of the iteration. Most edges are not fully evaluated with the clustering produced as output. They are typically evaluated with clustering obtained early in the computation. Still, we prove that when Algorithm 3 terminates successfully its output is the same as that of Algorithm 2.

**Theorem 2.** *If ties in Algorithm 3 are resolved in the same way as in Algorithm 2 then its clustering output is the same as the output of Algorithm 2.*

**Proof:** *The proof is by induction on the number of clusters.*

*It is easy to verify that for $k=n$ both algorithms return $n$ clusters, each consisting of a single node. Now suppose the theorem is true for $k$ and prove its correctness for $k-1$. From the inductive assumption both algorithms have the same $k$-clustering in the* **Union-Find** *data structure. Algorithm 2 selects the next edge as the one with the largest $h$ value computed from these volumes. We need to show that Algorithm 3 selects the same edge.*

*The edge selected by by Algorithm 3 is $e_1$ satisfying the condition $h_{1\_new} \geq h_{2\_old}$ (ineq 1). We need to prove $h_{1\_new} \geq h_{t\_new}$ (ineq 2), where $t$ ranges over all the edges in the heap, and the "new" indicates that they are computed from the volumes in the $k$-clustering. From the max-heap property we have: $h_{2\_old} \geq h_{t\_old}$ (ineq 3) for all $t$ in the heap, and from Proposition 1 we have $h_{t\_old} \geq h_{t\_new}$ (ineq 4) for all $t$ in the heap. The proof follows by observing that concatenating ineq 1,3,4 gives ineq 2.* □

**Complexity.** The space complexity is the amount of memory needed for the heap and for the **Union-Find**. The heap requires $O(m)$, and the **Union-Find** takes $O(n)$. Therefore, the amount of memory required by the algorithm is $O(m)$.

The running time is affected by the heap calculations and by the **Union-Find** calculations. The heap calculations take $O(xm \log m)$, where $xm$ is the number of **Extract_Max** calls. The **Union-Find** takes $O((n-k) \log n)$. Therefore the running time complexity is $O(xm \log m)$. We proceed to show that $x$ is bounded by $\log n$. Experiments described in Section 3 show that $x$ is almost always a small constant.

**Lemma 1.** *The number of* **Extract_Max** *calls in Algorithm 3 is at most $\log_2 n$.*

**Proof sketch:** *We first observe that the worst case is the complete graph with equal weights. Consider the special case where the graph is partitioned into $c$ clusters of equal size. It can be shown that after an additional selection of $c/2$ edges the graph will be partitioned into $c/2$ clusters of approximately the same size. The worst case is when all $c/2$ clusters are of exactly the same size. It can further be shown that while these edges were selected, the number of edges that were extracted and then put back into the heap is: $\alpha(c) = n^2(c-2)/(2c) \leq n^2(n-2)/(2n) = n(n-2)/2$. Therefore, the total number of extractions in this worst case is:*

$$\alpha(n) + \alpha(n/2) + \alpha(n/4) + \ldots + \alpha(2) \leq (\log_2 n)n(n-2)/2$$

*This gives:*

$$x = \text{number of extractions}/m \leq (\log_2 n)n(n-2)/n(n-1) < \log_2 n \quad □$$

Our experiments show that the worst case considered in the proof of Lemma 1 significantly overestimates typical values obtained on real data. See Section 3.3.

## 2.2 MULTI-VIEW CLUSTERING

As discussed in Section 1.2.2 multi-view clustering computes clustering of several graphs sharing the same set of nodes. The most straightforward clustering criteria are obtained from sum / or average of single view criteria. Specifically, the multi-view equivalent of gcut would be:

mvgcut$=\frac{1}{2}\sum_{z=1}^{v}\sum_{t=1}^{k}\frac{C_t^z}{V_t^z}$. In this formula $v$ is the number of views, and $C_t^z, V_t^z$ are the cut and the volume of cluster $t$ in view $z$. The equivalent of Equation (2) is given by:

$$\text{mvgcut}=\frac{1}{2}\sum_{z=1}^{v}\sum_{t=1}^{k}\frac{C_t^z}{V_t^z}=\sum_{z=1}^{v}\sum_{(i,j,w^z)}h(i,j,w^z)=\sum_{(i,j)}\sum_{z=1}^{v}h(i,j,w^z)=\sum_{(i,j)}h_{\text{mv}}(i,j) \quad (3)$$

$$h_{\text{mv}}(i,j)=\sum_{z=1}^{v}h(i,j,w^z), \quad h(i,j,w^z) \text{ is defined in Equation (2).} \quad (4)$$

Algorithm 3 can be easily modified for handling multi-view clustering. The differences are as follows.

- **Input:** $v$ graphs on the same set of vertices.
- **Data Structures:** The algorithm maintains one heap and $v$ Union-Find instances.
- **Edge $h$ value:** The formula for the $h$ value of an edge is given by Equation (4).

**Complexity.** The space complexity is the amount of memory needed for the heap and for $v$ **Union-Find** instances. The heap requires $O(mv)$, and each the **Union-Find** takes $O(n)$. Therefore, the amount of memory required by the algorithm is $O(mv)$.

The running time is affected by the heap calculations, and by the **Union-Find** calculations. The heap calculations take $O(xm \log m)$ and the **Union-Find** takes $O(v(n-k) \log n)$. Since $x$ is taken as a small constant, the running time complexity is $O(m \log m + v(n-k) \log n)$.

## 2.3 RANDOMIZATION

In this section we describe a randomized implementation of Algorithm 3. The randomized algorithm is more accurate than the greedy version, and can be used to compute clustering according to various criteria. The main idea is straightforward: instead of using the "$h$" values to select an edge, select an edge at random with probability proportional to $h$. A direct implementation of this idea requires computing the $h$ values for all edges, which we consider too slow. Instead, we describe a solution that can be implemented with a minor change to Algorithm 3.

The key is the classical algorithm of Eframidis & Spirakis (2006). Consider a subset of $m$ elements and their associated nonnegative $h$ values. Let $h_t$ be the value associated with element $t$.

Eframidis and Spirakis show that a random selection of an elements with probability proportional to $h$ can be achieved by selecting the element with the largest rand_$h$ value, defined by: rand_$h_t = r_t^{1/h_t}$, where $r_t$ is selected uniformly at random from the range $[0, 1]$. The implementation of this idea is shown as Algorithm 4. It uses the following definition:

$$\text{if } h(i,j,w)=0 \text{ then } h(i,j,w,r)=0, \quad \text{otherwise } h(i,j,w,r)=r^{1/h(i,j,w)}.$$
$$h(i,j,w) \text{ is computed from Equation (2).} \quad (5)$$

**Correctness.** The internal iteration in Algorithm 4 runs identical to Algorithm 3, but with rand_$h$, as computed in Equation (5), instead of $h$, as computed in Equation (2). The proof of Theorem 2 relied on the monotonicity of $h$. We need to show that rand_$h$ is also monotonic. To see this it is enough to notice that from Equation (5) $h(i,j,w,r)=r^{1/h(i,j,w)}$, so that $h(i,j,w,r)$ is monotonic in $h(i,j,w)$. We point out that this property holds because we keep the random value $r$ fixed for each edge during a random iteration. This is essential for the correctness of the randomized algorithm.

**Extensions:**

- Algorithm 4 can be generalized to the multi-view case using the same procedure as the extension of Algorithm 2 to the multi-view case. We skip the details.
- Instead of returning the "single best" solution as the output of Algorithm 4 one can retain several top solutions.
- Different criteria for evaluating the output and determining top solutions.

---

**Algorithm 4:** Randomized fast graph clustering

---

**Input:** The $m$ edges $(i, j, w_{ij})$. The normalization values $p_1, \ldots, p_n$. (See Definition 1.)

$\qquad$ $k$ - the desired number of clusters.

$\qquad$ riters - number of random iterations.

$\qquad$ Criterion- the clustering criterion (e.g., ncut)

**Random iterations:** iterate for $s = 1, \ldots,$ riters:

$\qquad$ **1.** Select $m$ random values $r_1, \ldots, r_m$ uniformly from $[0, 1]$.

$\qquad$ **2.** Create the initial clustering $A_i=\{i\}$, $V_i=p_i$, for $i=1, \ldots, n$.

$\qquad\qquad$ Insert these subsets and their volumes into a **Union-Find**.

$\qquad$ **3.** For each edge $s$ compute the value $\mathsf{rand\_}h(i, j, w, r_s)$ according to Equation (5).

$\qquad\qquad$ **Make_Heap** from the edges with their corresponding $\mathsf{rand\_}h$ values.

$\qquad$ **Iteration:** Repeat until $n-k$ edges are selected:

$\qquad\qquad$ **1.** **Extract_Max** to obtain the edge $e_1=(i_1, j_1, w_1)$, with the value $h_{1\_\mathrm{old}}$

$\qquad\qquad$ **2.** Update the value of $h_{1\_\mathrm{old}}$ using Equation (5) with the most current clustering in

$\qquad\qquad\qquad$ **Union-Find**. Denote the new value $h_{1\_\mathrm{new}}$.

$\qquad\qquad$ **3.** **View_Max** on the heap (after $e_1$ extraction) to obtain the edge $e_2$ with value $h_{2\_\mathrm{old}}$.

$\qquad\qquad$ **4.** If $h_{1\_\mathrm{new}} \geq h_{2\_\mathrm{old}}$ perform **Union**$(A(i_1), A(j_1))$.

$\qquad\qquad\qquad$ Otherwise **Insert_Element**$(e_1, h_{1\_\mathrm{new}})$.

$\qquad$ **End of random iteration:** Retain the subsets in **Union-Find** with their Criterion value.

**Output:** The retained solution with the best criterion value.

---

## 3 EXPERIMENTAL RESULTS

**Clustering criteria.** As discussed in Section 1.1 classical spectral clustering optimizes normalized cuts, which leads to ncut and rcut. The explicit formula as given in Definition 2 is: $\mathsf{gcut}=\frac{1}{2}\sum_{t=1}^{k}\frac{C_t}{V_t}$. Thus, minimizing gcut gives a minimization on the average of $\frac{C_t}{V_t}$, but no guarantees on any particular cluster. An alternative is the generalized Cheeger criterion given by:

$$\mathsf{Cheeger} = \max_t \frac{C_t}{V_t}. \tag{6}$$

See, e.g. Lee et al. (2014). As in the case of generalized cuts this formula specifies two criteria, depending on the normalization. We refer to them as CheegerNcut and CheegerRcut. In our experiments we use CheegerNcut which appears to be the most common.

As pointed out in Equation (2) the formula for gcut can also be written as: $\mathsf{gcut}=\sum_{\text{cut edges }(i,j,w)} w(1/V(i)+1/V(j))$. Therefore, gcut can also be viewed as minimizing the average $w(1/V(i)+1/V(j))$ over all cut edges. As in the Cheeger case, minimizing the average does not guarantee anything about a single edge. We propose the following measure that does give such guarantees:

$$\mathsf{linfcut} = \max_{\text{cut edges }(i,j,w)} w(1/V(i)+1/V(j)) \tag{7}$$

As in the other cases this gives two criteria depending depending on the normalization. We refer to them as linfcutNcut and linfcutRcut. In our experiments we use linfcutNcut.

**Evaluation criteria.** There are two families of evaluation criteria for the result of clustering. The first is based on external criteria, and requires ground truth, and the second is based on internal criteria, that can be determined from the data itself.

### 3.1 CLUSTERING QUALITY AND SPEED

**Single view, internal criterion.** This experiment was performed with the following 4 algorithms: **Classical** is a direct implementation of Algorithm 1. **FastSimple** is the Macgregor algorithm. **Heap** is the implementation of Algorithm 3. **Heap20** is the implementation of Algorithm 4 with 20 random iterations. The graph was computed from an image obtained from the Coil dataset(see Nene et al. (1996)).

| Algorithm | k = 2 | k = 3 | k = 4 | k = 5 | k = 6 | k = 7 | k = 8 | k = 9 |
|---|---|---|---|---|---|---|---|---|
| | | | | ncut values | | | | |
| Classical | 0.0039 | 0.0077 | 0.0112 | **0.0133** | 0.0247 | 0.0332 | **0.0372** | **0.0526** |
| FastSimple | 0.0421 | 0.0956 | 0.1601 | 0.2202 | 0.2739 | 0.3254 | 0.4701 | 0.4697 |
| Heap | 0.0031 | 0.0071 | 0.0102 | 0.0148 | **0.0193** | **0.0295** | 0.0390 | 0.0555 |
| Heap20 | **0.0021** | **0.0059** | **0.0088** | 0.0140 | **0.0193** | **0.0295** | 0.0390 | 0.0555 |
| | | | | running time | | | | |
| Classical | 12.7011 | 9.3243 | 7.4482 | 6.4024 | 6.3571 | 6.0650 | 5.4394 | 5.4563 |
| FastSimple | **0.1666** | **0.2054** | **0.2055** | **0.2146** | **0.2020** | **0.2163** | **0.2033** | **0.2409** |
| Heap | 0.286 | 0.277 | 0.285 | 0.282 | 0.261 | 0.272 | 0.261 | 0.247 |
| Heap20 | 0.837 | 0.811 | 0.832 | 0.86 | 0.829 | 0.814 | 0.824 | 0.828 |

Figure 1: Clustering experiment with obj43 from the COIL dataset

Table 2: Clustering experiment, external criteria, single view

| criterion | Algorithm | k = 2 | k = 3 | k = 4 | k = 5 | k = 6 | k = 7 | k = 8 |
|---|---|---|---|---|---|---|---|---|
| | Classical | **1** | **0.5135** | **0.5757** | **0.5481** | **0.5878** | **0.5476** | **0.4388** |
| ARI | FastSimple | 0.4748 | 0.06144 | 0.0966 | 0.0801 | 0.053 | 0.0465 | 0.0585 |
| | Heap | **1** | 0.5082 | 0.4579 | 0.4918 | 0.512 | 0.4807 | 0.5239 |
| | Classical | **1** | 0.5787 | **0.6248** | **0.6079** | **0.6404** | **0.6272** | 0.5712 |
| NMI | FastSimple | 0.3942 | 0.0701 | 0.1313 | 0.1292 | 0.1026 | 0.0970 | 0.1305 |
| | Heap | **1** | **0.5890** | 0.5390 | 0.5630 | 0.6130 | 0.5450 | **0.6080** |

The experimental results are shown in Figure 1. The top performers in terms of ncut values are clearly "Heap20" and "Classical". The results of "FastSimple" are quite bad. The running time shows a clear advantage of "FastSimple", with our "Heap" algorithm coming second.

**Single view, external criterion.** For this experiment we use the "Multiple Features" dataset from the UC Irvine Repository. The ground truth is available, and the measures that we use are the ARI and the NMI. The three algorithms that were compared are the Classical, the FastSimple of Macgregor, and Algorithm 3 that we call the "Heap". The results are shown in Table 2. Here the results of our algorithm are clearly below "Classical", but clearly superior to "FastSimple".

**Multi-view.** Experiments with multi-view spectral clustering are shown in Table 3. The comparison is with Kumar et al. (2011) that we call CoReg, and with Kumar & III (2011) that we call CoTrain. The graph was generated from a small image taken from the SF-MASK dataset. The results show that our algorithm is competitive in terms of quality.

## 3.2 HUMAN EVALUATION OF CLUSTERING CRITERIA

Examining the results of clustering on many images we come to the conclusion that in most cases the LinfCut criterion is superior to both Ncut and Cheeger. We were not able to distinguish between Ncut and Cheeger based on quality. One example is shown below:

Table 3: Multi-view clustering experiment using ncut

| Algorithm | k = 2 | k = 3 | k = 4 | k = 5 | k = 6 | k = 7 | k = 8 | k = 9 |
|---|---|---|---|---|---|---|---|---|
| CoTrain | 0.0014 | 0.0035 | 0.0110 | 0.0229 | 0.0265 | 0.0925 | 0.0611 | 0.0903 |
| CoReg | 0.0014 | **0.0022** | **0.0039** | 0.0381 | 0.0527 | 0.0626 | 0.0741 | 0.1146 |
| Heap | **0.0005** | 0.0031 | 0.0046 | **0.0092** | **0.0141** | **0.0194** | **0.0307** | **0.0405** |

Table 4: Amortized number of extractions

| Data | Algorithm | k = 3 | k = 4 | k = 5 | k = 6 | k = 7 | k = 8 | k = 9 |
|---|---|---|---|---|---|---|---|---|
| medium image | Single View | 4.65 | 4.64 | 4.64 | 4.63 | 4.63 | 4.61 | 4.59 |
| | Multi View | 3.68 | 3.67 | 3.66 | 3.66 | 3.65 | 3.65 | 3.65 |
| | $n$ | 16384 | 16384 | 16384 | 16384 | 16384 | 16384 | 16384 |
| | $\log_2 n$ | 14.00 | 14.00 | 14.00 | 14.00 | 14.00 | 14.00 | 14.00 |
| small image | Single View | 3.36 | 3.32 | 3.3 | 3.25 | 3.21 | 3.19 | 3.18 |
| | Multi View | 2.46 | 2.46 | 2.43 | 2.41 | 2.39 | 2.38 | 2.36 |
| | $n$ | 690 | 690 | 690 | 690 | 690 | 690 | 690 |
| | $\log_2 n$ | 9.43 | 9.43 | 9.43 | 9.43 | 9.43 | 9.43 | 9.43 |
| UCI | Single View | 6.33 | 6.34 | 6.47 | 6.58 | 6.61 | 6.62 | 6.59 |
| | Multi View | 5.48 | 6.09 | 6.32 | 6.4 | 6.48 | 6.42 | 6.44 |
| | $n$ | 600 | 800 | 1000 | 1200 | 1400 | 1600 | 1800 |
| | $\log_2 n$ | 9.22 | 9.64 | 9.96 | 10.23 | 10.45 | 10.64 | 10.81 |

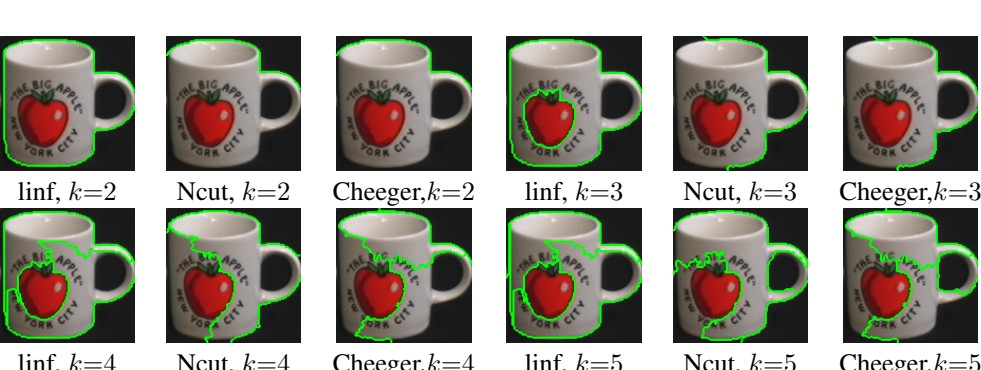

linf, $k=2$    Ncut, $k=2$    Cheeger, $k=2$    linf, $k=3$    Ncut, $k=3$    Cheeger, $k=3$

linf, $k=4$    Ncut, $k=4$    Cheeger, $k=4$    linf, $k=5$    Ncut, $k=5$    Cheeger, $k=5$

## 3.3 THE NUMBER OF EXTRACTIONS

Lemma 1 showed that the number of amortized extractions during the run of the algorithm is bounded by $\log n$. Table 4 shows the values obtained when the algorithm is applied to real data. Observe that typically the amortized value is significantly smaller than $\log_2 n$.

## 4 CONCLUDING REMARKS

Most spectral clustering algorithms are developed as variants of Algorithm 1. Our approach is different. We take a traditional computer science approach, looking for a solution by a greedy-search algorithm and improving it by randomization. With the exception of Macgregor's algorithm, our algorithm is the fastest. Macgregor's algorithm is sometimes faster, but typically less accurate.

The speed of our algorithm allows us to run a randomized variant, and evaluate clustering criteria that were not previously evaluated experimentally. While there is theoretical work on the importance of the Cheeger criterion, we are not aware of any experimental comparison between it and other criteria. The reason might be that there were no sufficiently fast algorithms for computing clustering with the Cheeger criterion. We suggest the linfcut criterion, which appears better to a human observer.

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

# A    APPENDIX

