# OpenReview forum: "Very Fast Graph Clustering for Single and Multiple Views"
_ICLR.cc/2025/Conference — ICLR 2025 Conference Withdrawn Submission_

### Official Review · Reviewer_sX5R · 2024-10-25

**Soundness:** 2
**Presentation:** 2
**Contribution:** 2
**Rating:** 3
**Confidence:** 3

**Summary:**

The paper presents a fast, greedy algorithm designed to produce an approximate solution to the minimum (normalized) cut problem in graphs. The algorithm is adaptable to various cut types (e.g., ratio cut, normalized cut) and can handle multilayer (multiplex) networks. Additionally, the authors introduce a randomized variant of their algorithm and assess its performance on several real-world datasets.

**Strengths:**

The proposed algorithm relies on a straightforward, greedy approach, which allows it to adapt easily to diverse scenarios and objective functions. This adaptability is a notable strength, as it enables the algorithm to be employed across a range of graph clustering tasks with minimal adjustments.

**Weaknesses:**

* The numerical evaluation is limited in scope. Experiments are conducted on a very small number of datasets, including only one with a ground-truth cluster and a single image from COIL. This limited evaluation does not suffice to draw meaningful conclusions about the algorithm’s general performance. To better assess its effectiveness, experiments on both synthetic and more extensive real-world datasets are required. For example, testing on standard graph clustering benchmarks like the SNAP datasets or generating synthetic graphs with known cluster structures of varying sizes and densities.

* The datasets used in the current experiments are small. Testing on larger graphs is necessary to conclude the algorithm's empirical runtime, especially as the paper claims scalability as a key benefit. For example, testing on graphs with 10^4, 10^5, 10^6, and more nodes would show how runtime scales with graph size.

* The usefulness of the proposed “FastSimple” algorithm becomes more relevant for cases where $k$ (the number of clusters) is large, given that it only uses $\log⁡ k$ vectors rather than computing $k$ eigenvectors. Comparisons with small $k$ values may, therefore, provide misleading insights into the algorithm’s efficiency relative to traditional methods.
For example, on 'Multiple Features' data set, it seems the correct number of clusters is $k=2$. Hence, using 'FastSimple' instead of 'Classical' is totally useless. I believe the difference in running time is simply due to a better implementation by the authors of FastSimple (MacGregor et al.) rather than an intrinsic advantage of FastSimple.

* The section on related works focuses heavily on spectral methods, but other methods have been proposed to solve the min-bisection problem (tracing back to Lin–Kernighan approaches), and those methods need to be discussed as they are related to the authors' proposal. Moreover, the paper [1] already shows that it is possible to partition a well-clustered graph in nearly linear time. Finally, ref [2] shows that spectral analysis can be carried out over extremely large graphs.

Missing references:

[1] Richard Peng, He Sun, and Luca Zanetti. "Partitioning well-clustered graphs: Spectral clustering works!." Conference on learning theory. PMLR, 2015.

[2] Kang, U., Meeder, B., & Faloutsos, C. (2011). Spectral analysis for billion-scale graphs: Discoveries and implementation. In Advances in Knowledge Discovery and Data Mining: 15th Pacific-Asia Conference, PAKDD 2011, Shenzhen, China, May 24-27, 2011, Proceedings, Part II, 13-25. Springer Berlin Heidelberg.

**Questions:**

* The numerical section mentions that the linfcut is better than the cut. Would it be possible to directly optimize for the linfcut instead of the cut? Are there specific advantages or potential issues in directly optimizing the linfcut compared to the (ratio or normalized) cut?


Details and Minor points

* Section 1.2.1:

 "to infer the graph from the data": Does this imply the graph is unobserved? If so, what form does the data take?

"with steps 1,2 of the algorithm": Which specific algorithm is being referenced here?

The two paragraphs on deep learning approaches appear tangential to the discussion.

"all others can be viewed as modifications of Algorithm 1": This statement applies only to the methods listed. Other types of algorithms exist, tracing back to Lin–Kernighan approaches.

"the expected time is": Expected time with respect to which randomness? The randomized nature of the algorithm should be introduced earlier.

"to he original": “he” should be “the.”

* Section 1.3: "running time is faster than a single iteration of k-means": Clarification needed—are the authors referencing Lloyd’s algorithm? Note that faster algorithms exist for approximate k-means clustering (such as mini-batch k-means).

* Step 3 of Algorithm 2: "replacing then" should be corrected to "replacing them."

* Page 7: "different criteria for evaluating…" is unclear and should be elaborated.

* Page 9: Additional information on the Multiple Features data set (e.g., number of vertices, clusters, edges) and the COIL image (e.g., graph construction methods, such as k-nearest neighbors and edge weighting) would be beneficial.

* Conclusion: The final paragraph is somewhat unclear and would benefit from further clarification.

---

### Official Review · Reviewer_CoAM · 2024-10-27

**Soundness:** 2
**Presentation:** 3
**Contribution:** 1
**Rating:** 3
**Confidence:** 4

**Summary:**

This work presents a very fast algorithm for graph-based clustering. It produces clustering assignments by recursively merging the clusters with the largest inter-cluster similarity. The authors also provided a fast implementation using a binary heap and an extension to multi-view clustering.

**Strengths:**

* The paper is well-written and easy to follow
* The convergence and complexity of the proposed algorithm are analyzed in detail

**Weaknesses:**

* Recursively merging clusters is not new but a widely adopted idea in hierarchical clustering, and this work is very similar to [1]. The core ideas (Algorithm 2) are both to start with singleton clusters and then merge the most similar pairs in each iteration, except that [1] is specialized for N-Cut only while this work considers a generalized graph-cut problem.
* Following Algorithm 2, this work introduces a fast implementation, Algorithm 3, which leverages the max-heap data structure to speed up max-value maintenance. I'm unsure if the same acceleration appears in previous works, but it's not an important academic improvement from my point of view because it appears in any data structure textbooks. Finally, Algorithm 4 is obtained by applying an **existing** random sampling technique to Algorithm 3. In summary, this work can be considered primarily an engineering achievement to [1], with limited academic novelty.
* The experimental evaluation is insufficiently comprehensive. The datasets used are outdated, small, and lack complexity, with the largest dataset containing only 16384 samples, as indicated in Table 4. This size is notably smaller than the MNIST dataset. Aimed at demonstrating the efficiency, datasets sized at 10k~100k are expected.
* There are essentially no competitors across single-view experiments. Since several spectral clustering algorithms have been surveyed in Table 1, they can also serve as competitors of the proposed algorithm. Also, the multi-view competitors were over a decade old. Given two recent multi-view surveys cited in this paper, I suggest following their settings to evaluate multi-view clustering performance.

[1] GANC: Greedy agglomerative normalized cut for graph clustering. Pattern Recognition, 2012

**Questions:**

* Many real-world graphs are 01-valued and don't come with edge weights, can the proposed algorithm handle them?
* Recursively merging clusters may result in imbalanced assignments, i.e. producing extremely large or small clusters, how does the proposed algorithm prevent this?

---

### Official Review · Reviewer_quDV · 2024-10-29

**Soundness:** 2
**Presentation:** 3
**Contribution:** 2
**Rating:** 3
**Confidence:** 4

**Summary:**

This paper studies fast spectral clustering, and proposes a new algorithm for optimizing the Ratio Cut and Normalised Cut objectives without using the spectrum of the graph matrices. The proposed algorithm is essentially a greedy agglomerative clustering algorithm based on the normalised cut objective function, and can be efficiently implemented using heap data structures, giving a total running time of $O(m \log(m))$ (independent of $k$).

The paper further provides some experimental evidence that the new algorithm performs well in comparison with alternative spectral clustering algorithms, and proposes a new objective function, linfcut, to be used in place of normalised cut or Cheeger cut objectives.

**Strengths:**

This paper is clearly written, and provides a good overview of the state-of-the-art for spectral clustering algorithms. The new proposed algorithm is quite a nice, natural idea and it is well motivated by the normalised cut objective function. The proposed algorithm is original and an interesting contribution to the spectral clustering literature.

**Weaknesses:**

This paper has a number of issues which limits its impact in its current form.

1. The theoretical results are not proved in the paper, nor in the supplementary material. The running time guarantee in Lemma 1 is really the central claim of the paper - there should be a full proof of this claim.
2. The introduction claims that the new algorithm minimizes the same criteria as classical spectral clustering algorithms. There is in fact no guarantee that the proposed algorithm minimises any particular objective function, while classical algorithms minimise a relaxed normalised cut objective.
3. Given that the proposed algorithm is essentially heuristic, it should be justified with extensive experimental results. The current experiments are on only a single image from one dataset.
4. There is no code provided to reproduce the experimental results. Again, for a heuristic algorithm, reproducibility of the results is very important.

The paper could be significantly improved by firming up the theoretical analysis of the algorithm and running much more extensive experiments.

**Questions:**

1. The introduction claims that the expected running time of the new algorithm is $O(m log(n))$ - is this proved in the paper?
2. The randomised algorithm has no accompanying theoretical guarantees. I am concerned that by keeping the values of $r$ fixed between iterations, the edges are not sampled independently.
3. On page 7 is the claim that the randomized algorithm is more accurate than the greedy algorithm - is this a theoretical or empirical claim?

I also have a more general question: given that the algorithm is an agglomerative clustering algorithm, is there a connection to the literature on hierarchical clustering, and optimisation algorithms for Dasgupta's cost function [1]?

[1] S. Dasgupta. A cost function for similarity-based hierarchical clustering. 2015

---

### Official Review · Reviewer_TxcR · 2024-10-29

**Soundness:** 2
**Presentation:** 2
**Contribution:** 3
**Rating:** 3
**Confidence:** 3

**Summary:**

The authors present a graph clustering algorithm that is faster than the state-of-the-art when the graph is sparse and k is big.
They also have results for multi-view clustering. To achieve the strong results, the authors rely heavily on efficient data-structures. I'm somewhat surprised that this runtime is not known, since it seems like a very important regime.

**Strengths:**

The runtime of the algorithm is very good. The regime the authors operate in is a very interesting one. Give more examples on why this regime is interesting.

The data structures used are quite clever.

The proof sketches are very useful.

**Weaknesses:**

The biggest weakness is that there are only proof sketches. The algorithm is quite complex and without solid proofs, I cannot accept the paper. Note that I'd rate the paper much higher if there had been proper proofs. It seems to be that the appendix is empty and the supp material only contains the code.

My second biggest concern is that the related work section is a little short. Is it clear that the Label Propagation Algorithm or Louvain have a worse runtime? At the very least they should be mentioned.


Code has no comments.

Lemma 1, why introduce x?

Some parts of the paper feel very unnatural, e.g. (5)

The "human evaluation" is very strange and lacks scientific rigor.

Table 3 and 4 are not very clear.

The related work is rather short.

**Questions:**

Why are the examples in the paper for clustering cups (images) rather than graphs? Aren't there more natural graphs one could use?

What are the different colors (red\green) for?

---

### Official Review · Reviewer_yEb9 · 2024-10-30

**Soundness:** 2
**Presentation:** 2
**Contribution:** 1
**Rating:** 3
**Confidence:** 5

**Summary:**

The paper studies faster algorithms for spectral clustering for single and multiple views. Given an input graph $G$ of $n$ vertices and $m$ edges, the designed algorithm runs in $O(m\log m)$ time in the single view setting, and $O(m\log m +v(n-k)\log n)$ in the multiple view setting.  The paper conducts experimental studies comparing their designed algorithms with the others in the literature.

**Strengths:**

One strength of the paper is to show that efficient data structures can be used to speed up the running time of clustering algorithms.

**Weaknesses:**

The paper lacks the theoretical analysis of their designed algorithm, and this is one of the major weakness of the paper. Specifically, assuming $S_1,\ldots, S_k$ are the optimal clusters defined under a certain metric, it's unclear how good the algorithm's output approximates $S_1,\ldots, S_k$.

The other major weakness of the paper is the lack of comparison with more related work on spectral clustering. For example, it is known that spectral clustering can be implemented in nearly-linear time with theoretical guarantee (See: Peng et al., SICOMP'17). Compared with the result by Peng et al., the running time improvement of the current algorithm is just a poly-logarithmic factor; however, the current algorithm doesn't have any theoretical guarantee of the output clustering.

Given the two major downsides, in my point of view the paper is clearly below the standard of ICLR.

**Questions:**

I would welcome a detailed comparison between the paper and Peng et al. 2017.

---

### Official Review · Reviewer_p4dp · 2024-10-31

**Soundness:** 2
**Presentation:** 2
**Contribution:** 1
**Rating:** 3
**Confidence:** 5

**Summary:**

This paper introduces a novel, fast algorithm for graph clustering that bypasses the traditional reliance on spectral techniques, opting instead for a greedy agglomerative approach; it starts with each node as a separate cluster and then merging pairs based on an edge weight criterion until the desired number of clusters is achieved. To speedup the algorithm, union-find and max-head datastructures are used + a randomization scheme. This provides improvement over classical methods by bypassing expensive eigenvector computations. The algorithms can also be applied to the multi-view setting.

The paper further introduces a  new "LinfCut" criterion, as an alternative to normalized cut and Cheeger cut criteria, and it is claimed that this criterion is superior as it "makes more sense'' to the human observer. Experiments demonstrate that the algorithm achieves comparable results to spectral clustering on several datasets while operating faster, with results reported for internal clustering quality (ncut) and external criteria, such as adjusted Rand index (ARI) and normalized mutual information (NMI).

**Note:** For the rest of the review, any new references are indicated with a number (e.g. [1]). These references are listed at the end of the Weaknesses section. For existing references I use authors+year

**Strengths:**

**S1.)** Previous work on finding clusters with low conductance have focused on using spectral approaches. This is mainly due to the early success of spectral clustering (Shi & Malik, 2000; von Luxburg, 2007), and subsequent theoretical work closely relating the conductance to the spectrum of the Laplacian matrix through the Higher-Order Cheeger inequality (Lee et al., 2014). This paper takes an entirely different approach, and proposes relatively simple algorithms for finding clusters with low conductance with a greedy merging criterion. This is rather surprising, and it could lead to exciting new work in the area.

**S2.)** The naive implementation of the greedy algorithm can be quite slow; this paper proposes several clever ways to speed up this algorithm by (i) using union-find and max-heaps datastructures; (ii) some clever checks to avoid recomputation (e.g. line 4 of alg 3); and (iii) randomisation techniques.

**Weaknesses:**

**W1.)** There are no theoretical guarantees on the returned clusters. The focus and main interest of this work is running time for (spectral) clustering. The paper states that this is also the focus of most current research. I do not believe that this is accurate. Several recent works have focussed on improving the approximation guarantee of spectral clustering (e.g. [3]). Of the works that do improve running times, there is a guarantee that the returned approximation guarantee is as good or slightly worse than the previous state-of-the-art, e.g. Macgregor (2023) & Chen et al. (2023). The lack of any theoretical approximation guarantees in this paper is therefore a downside. I do understand that obtaining these guarantees is likely difficult due to the greedy approach of the algorithm. However, even proving a result in a simplified regime, such as for $d$-regular graphs with $k=2$ clusters (and potentially on the average normalized cut instead of the Cheeger-based conductance) would significantly improve the impact of this work.


**W2.)** There are several key references to the literature that are missing, and the lack of these weakens the contribution of the paper.

**a.)** The greedy heuristic is essentially a form of (hierarchical) agglomerative clustering. These types of algorithms are _widely_ studied, but this work contains no references to them. See [2] for an overview. Furthermore, techniques such as union-find and heaps are very common in this area, which diminishes the novelty of this work.

**b.)** A very similar greedy heuristic was proposed in [1], where clusters are merged based on the maximal improvement in normalized cut; this submission differs by merging clusters based on the maximal edge improvement using $h(i, j, w_{ij})$. Still, in my opinion the ideas are conceptually too close for this work to be considered novel. Indeed, using similar techniques as this paper, [1, section 4.4] achieves an almost identical running time to this work.



**W3.)** Given that the focus of this work is proposing good heuristics (algorithm 2, 3 and 4), without any theoretical approximation guarantees, it would have been nice to see an extensive experimental evaluation. However, there are several issues with the experimental section.

**a.)** The only evaluated datasets are: (i) image obj43 from the COIL dataset; (ii) the "Multiple Features'' dataset from the UC Irvine Repository; and (iii) a small image from the SF-MASK dataset. This is not enough the evaluate the proposed algorithm: It is not clear if these single images are hand-picked to provide the best results compared to the baselines. Second, given that one of the main contributions of this paper is a very fast algorithm for finding clusters with low conductance, I would expect to see larger datasets evaluated. This does not even need to be done on real-world instances, and it could be done by evaluating the algorithm on stochastic block models of increasing size, thereby showing the running time asymptotics.

**b.)** In section 3.2, the paper claims that the newly proposed LinfCut is superior to both Ncut and Cheeger. The argument for this is that "examining the results of clustering on many images we come to the conclusion that in most cases the LinfCut criterion is superior to both Ncut and Cheeger''. This is a bold claim, as Ncut and Cheeger have been used in the field for decades. However, the claim is backed up by showing a _single_ example where the algorithm returns better clusters - this is unscientific. Again, this could be a hand-picked image from the dataset that best showcases the algorithm. To make this claim, I would either expect a randomized control trial on humans. This could consist of randomly showing images to real-life participants who could then rate the images (perhaps using an elo system to determine the best algorithm). Another non-human involved way would be to evaluate the metric on ground truth segmentations, of well-known image segmentation datasets (e.g. BSDS, or more images from COIL and SF-MASK), to see what criterion best matches the clusters. Calling something a "human evaluation of clustering criteria'' and then showing a result of a single image is below the bar of acceptance.

**c.)** Some key experimental details are missing for reproducibility, and I've listed these as questions below.


**Minor points/typos**
- End of page 1 typo: Laplaian --> Laplacian.
- Point 1 end of section 1 typos: "An $O(m \log m)$(expected) clustering algorithms (...) '' --> "An $O(m \log m)$ (expected) clustering algorithm (...) ''
- 2nd sentence sec 1.3 typo: Bader(2022) --> Bader (2022)
- Appendix A section header is placed but there is no appendix. Can be deleted.
- At start section 2: in the definition of gcut, $w$ is given as an argument for $h(i, j, w)$, but later on $h(i, j, w_{ij})$ is used in the algorithm. Would be better if this is consistent (Same point holds for equation 4 in the multi-view setting).
- No mention of computing equipment was reported - reporting these would be good for reproducability (See Q4).
- Sentence after equation 7 typo: "two criteria depending depending''
- In section 3, the paper states "in our experiments we use CheegerNcut which appears to be the most common''. Adding references here to backup the claim would be useful.


**Additional references**:

[1] Tabatabaei, Seyed Salim, Mark Coates, and Michael Rabbat. "GANC: Greedy agglomerative normalized cut for graph clustering." Pattern Recognition 45.2 (2012): 831-84

[2] Müllner, Daniel. "Modern hierarchical, agglomerative clustering algorithms." arXiv preprint arXiv:1109.2378 (2011).

[3] Macgregor, Peter and Sun, He. "A Tighter Analysis of Spectral Clustering, and Beyond". ICML 2022.

**Questions:**

**Q1.)** What is $l$ in Table 1? Is it a parameter used in algorithms 2-6? For ease of readability, it would be good to state all the parameters in the table description, i.e., "$n$ denotes the number of vertices, $m$ the number of edges, $k$ the number of clusters and $l$ the ...".

**Q2.)** In the experiments, how exactly are the images turned into graphs? Is a Gaussian kernel used (if so, what parameter $\sigma$)?  What is the dimensionality of the images?

**Q3.)** For the Multiple Features dataset: how large is it? How many features per data point? How many ground truth clusters are there? The last question is particularly relevant for understanding the results of Table 2.

**Q4.)** What type of compute was used to obtain these experiments? E.g. CPU, RAM details.

---

### Note · Authors · 2024-11-13

I have read and agree with the venue's withdrawal policy on behalf of myself and my co-authors.